# Pilot study to explore the use of mobile spaced learning as a digital learning platform when teaching symptom management to undergraduate nursing students: SPLENdidS study

**Clare Mc Veigh** **\*⍟, Susan Carlisle⍟, Matt Birch⍟, Lindsay Ace⍟, Christine Oliver⍟, Helen Kerr⍟**

School of Nursing and Midwifery, Queen's University Belfast, Belfast, United Kingdom

⍟ These authors contributed equally to this work.
\* clare.mcveigh@qub.ac.uk

## Abstract

### Background

The management of patients' holistic symptom needs are often complex and challenging. The education needs of undergraduate nursing students must be optimally addressed to have a significant positive impact on patient care. Mobile spaced learning has been recognised as a form of online education which can provide a novel approach to delivering effective evidence based healthcare education to undergraduate students.

### Objective

The objective of this pilot study was to explore the experiences of undergraduate nursing students in a university setting, of using mobile spaced learning as a digital platform for symptom management education.

### Method

This pilot study used a mixed methods approach. Online spaced learning material, which utilised both case based scenarios and multiple choice questions, was delivered to first year undergraduate nursing students over a period of 2 weeks. Participants were then invited to participate in an online survey related to the usability of mobile spaced learning. A focus group was conducted to further explore the participants' views.

### Result

Findings conveyed that students viewed mobile spaced learning as an acceptable platform that enhanced both their learning and their ability to transfer knowledge into clinical practice.

**Data Availability Statement:** Data underlying the results presented in the study are available from a public repository at Queen's University Belfast: DOI

10.17034/3b8fecf0-585d-4543-b712-92f02f3d63d3.

**Funding:** CV received the funding from the Martha Mc Menamin Memorial fund. Funders did not play a role in the study design, data collection and analysis, decision to publish, or preparation of the manuscript.

**Competing interests:** The authors have declared that no competing interests exist.

## Conclusion

Implementation of a digital spaced learning intervention would be acceptable to undergraduate nursing students learning about holistic symptom management. Further research is needed to explore the feasibility of implementing this intervention within the undergraduate nursing curriculum, and also to explore the impact on long-term knowledge retention.

## Introduction

Online learning enables healthcare education to extend to a wider audience in a more accessible manner. However, one of the key challenges when promoting an online education environment is ensuring that the delivery method encourages participation of the learner and provides effective knowledge translation. Mobile spaced education, which is a learning analytics platform that promotes active learning [1], can provide a novel approach to optimising effective evidence based healthcare education. Spaced learning is achieved through delivering short clinical case based scenarios to students via email or a handheld mobile device [2]. A small number of case scenarios are delivered to the student every other day with relevant questions regarding the case. Each student's performance is recorded to allow for analysis, whilst students are provided with immediate and succinct feedback to the questions.

This online platform is built upon the psychological theory of learning that education that is 'spaced' and 'repeated over time' can deliver more efficient learning and improved retention compared to a single, and less interactive, distribution learning format [3]. Previous research amongst urology medical students, highlighted that mobile spaced education generated significant topic-specific knowledge and resulted in improvements in learning that were retained for at least two years [4]. Several randomised controlled trials (RCTs) have demonstrated that spaced learning is an evidence based pedagogy that can increase clinician knowledge, whilst also altering behaviours in the healthcare setting [1, 3, 5, 6]. In one RCT Kerfoot et al. [5] investigated the use of online-spaced learning to deliver educational content relating to hypertension management to clinicians in primary care. The results indicated that this specific educational intervention reduced the time taken for patients to achieve their target blood pressure range [5].

Within the nursing profession, an Australian study illuminated that mobile spaced education improved nurses self-perceived pain assessment and management skills within the oncology setting [2]. Phillips et al. [2] aimed to test the impact of an online spaced learning intervention on oncology nurses' assessment skills in relation to pain management. Self-perceived competencies in relation to pain assessment were measured amongst participants at three time points [2]. Phillips et al. additionally highlighted that the positive impact of the intervention was also displayed by an increase in the documentation of pain assessments within patients' charts. Mobile spaced learning offers the opportunity to deliver specialised clinical content in an on-line format that has the potential to improve practice [7].

Previous research has focused almost exclusively on the use of innovative methods of technology and education for patients, rather than the delivery of evidence-based information to healthcare professionals [8]. The management of patients' holistic symptom needs are often complex and challenging with a clear need for a strong and integrative approach to care in clinical practice. To achieve this, the education needs of undergraduate nursing students involved in holistic nursing care needs to be optimally addressed to have a significant impact on patient care. This pilot study explored the practicalities of implementing a mobile spaced

learning model to deliver symptom management education, to undergraduate nursing students. This study lends itself to further interventional research and provides in depth information regarding the implications of introducing this model into undergraduate nurse education. The next stage will focus on testing the feasibility of the intervention based on the results from the pilot study and also exploring further research to measure the impact of this educational methodology on patient outcomes.

## Methods and design

### Research aims

This study aimed to explore the practicalities of implementing a mobile spaced learning model to deliver symptom management education to undergraduate nursing students.

The objectives of this study were to:

1. Explore the usability and acceptability of a mobile spaced learning platform in providing symptom management education to first year undergraduate nursing students.

2. Determine if mobile spaced learning can be successfully implemented to deliver symptom management education to first year undergraduate nursing students.

3. Explore participants' perceptions of the impact of the spaced learning digital intervention on the development of knowledge in relation to symptom management.

### Research approach

This pilot study adopted a mixed methods approach to explore the practicalities of implementing the proposed digital learning model. The qualitative element was based on broad interpretivism. The philosophy of interpretivist research is that the researcher will interpret the world they investigate [9]. The aim of an interpretivist approach is to understand the world from the point of view of the participants in it rather than the world's explanation [10], it was therefore, consistent with the aims of the study.

### Ethical considerations

Ethical approval was granted from the School of Nursing and Midwifery Research Ethics Committee (Ref: 5. CMcVeigh 04.18.M5.V1). Prior to the commencement of the study all participants were made aware they could withdraw from the study at any point with no negative consequences. Participants were made aware prior to data collection that every endeavor would be taken to protect their privacy and identity throughout the duration and after the study is complete. In the reporting of results, pseudonyms are used to protect the identity of participants when verbatim quotations are used.

### Intervention

The online spaced learning material related to symptom management was embedded in a current module being delivered in the first year of the undergraduate BSc (Hons) Nursing programme at one University in the United Kingdom. Case based scenarios and multiple choice questions (MCQs) were developed by members of the research team, who teach the module involved and also have expertise in symptom management. Scenarios and questions developed were based on the learning outcomes of the module and the content that it delivers [11]. Fig 1 outlines the intervention and how this was delivered. On completion of the face-to-face learning on symptom management, three case based symptom management scenarios were

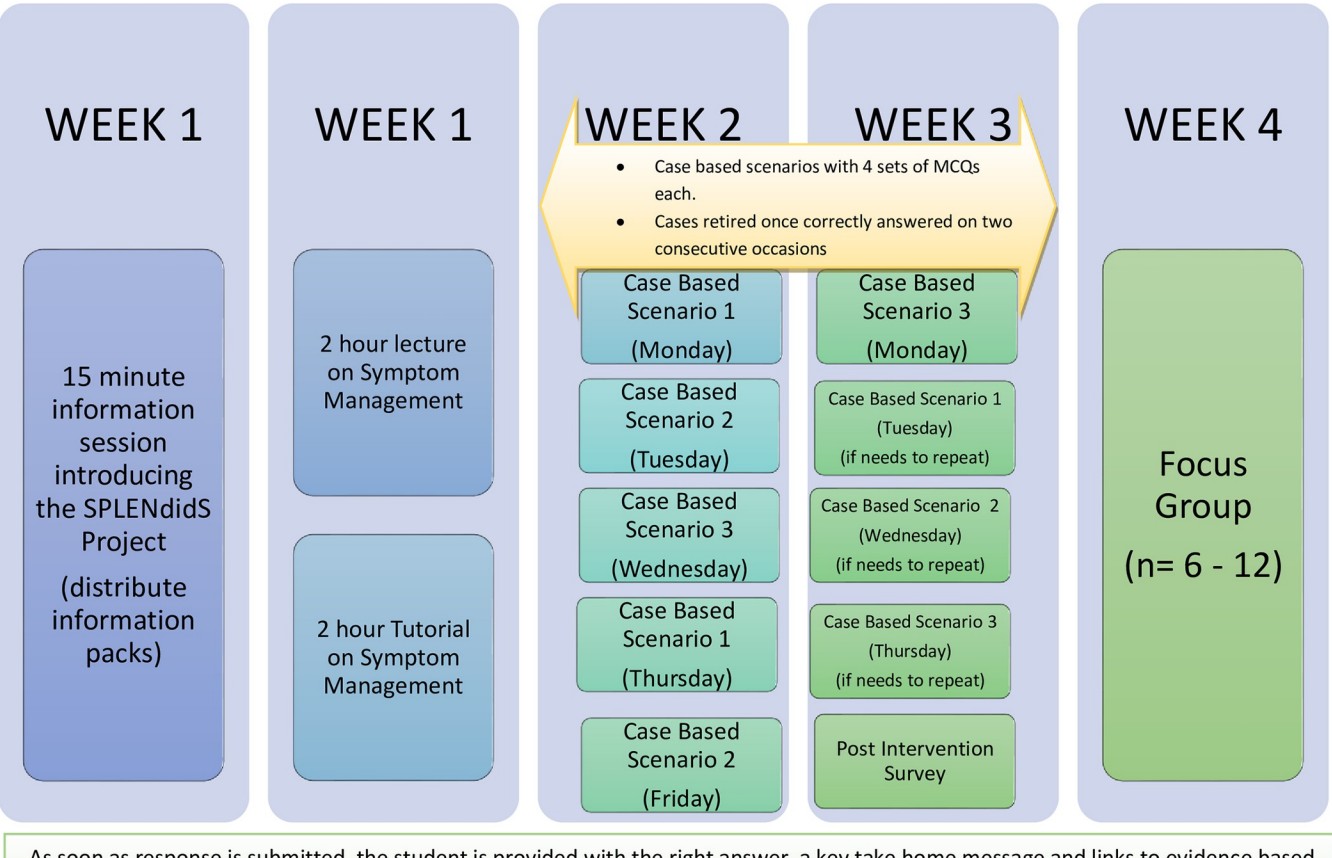

**Fig 1. Intervention.**

delivered directly to consenting participants' e-mails in a spaced, repeated, and tested format over 2 weeks. This could then be accessed on a desktop or mobile device. Each case scenario was tested using four sets of MCQs. Participants engaged with each case and completed the related questions in a location and time of their choice. The correct answer to each MCQ question was provided as soon as a response was submitted, providing participants with the answer, a key take home message and links to evidence based resources. If participants correctly answered all the MCQs for the case scenario on the first attempt, then this scenario was forwarded to them again in 72 hours. After they correctly answered all the MCQs for a scenario twice, they did not receive this scenario again. If a participant was unsuccessful on their first, second, or both attempts in correctly answering all MCQs for a scenario, they had a third opportunity to repeat this, 72 hours after their last attempt. Participants only received each scenario a maximum of three times. Responses to the MCQs were only used for the purposes of this research and not towards any summative assessment. Lecturers in the School of Nursing and Midwifery outside the research team did not have access to participant's individual responses.

## Sample and participants

Convenience sampling was used to recruit relevant participants into the study. This involved selecting participants who were able to be accessed conveniently and efficiently. By using

convenience sampling the researcher was able to accept any student who met the criteria and agreed to take part in the study as a participant [12]. Participants of the study were eligible if they were a first year undergraduate student on the BSc (Hons) Nursing programme, who were completing the Professional Nursing Values Module. There were 135 nursing students from one cohort of students who were invited to participate in the study. It was proposed that six to twelve participants would be recruited to participate in the focus group.

Recruitment commenced in September 2018. An information session was delivered to all students in week one of phase three. Information packs were also provided at the end of the information session containing an invitation letter, information sheet, and consent forms for both the study and questionnaire, and the focus group. Before the focus group commenced, verbal and written informed consent was obtained from each participant.

First year undergraduate nursing students (n = 135) were invited to take part and given information packs. As Fig 2 highlights, 20% (n = 27) of students consented to take part in the

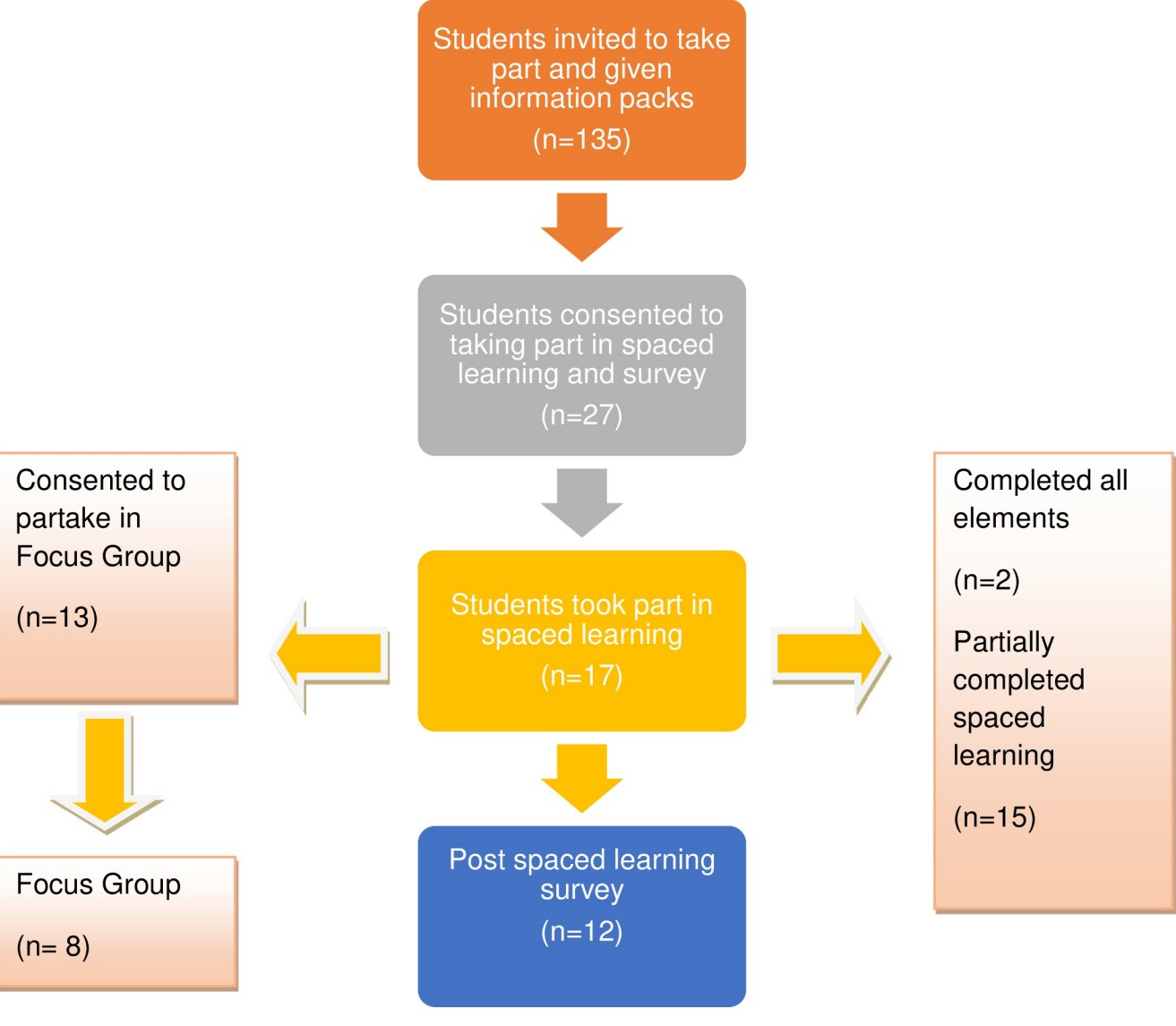

**Fig 2. Study recruitment and retention.**

study, however 10 did not participate leaving a total of 17 participants from the cohort of students (13%) who took part in the spaced learning. Seventy-one percent of these participants (n = 12) completed the online survey after the spaced learning was complete. For the focus group, 13 students consented and 8 of these took part (47% of participants). In relation to the spaced learning, 12% (n = 2) completed all required elements, and 88% (n = 15) only partially completed the spaced learning activities.

## Data collection

At the end of the two-week period of spaced learning (weeks two and three on Figure one), a report was generated via the digital platform to show participants performance. To evaluate the acceptability and usability of the spaced learning approach using mobile technology (S1 File), participants were then invited to participate in an online survey (S1 File) via survey monkey related to the usability and acceptability of spaced digital learning as a learning method within the university setting. Following this, a focus group (Table 1) was conducted with 8 participants to further explore their views on the acceptability and usability of the mobile spaced learning education approach. Focus groups were digitally recorded and transcribed verbatim.

## Analysis

Data from students' engagement in the MCQs using spaced learning was collated by the research team and the survey data was analysed using descriptive statistics. Qualitative data from the focus group was analyzed using a thematic content analysis approach as described by King and Horrocks [13]. To assist the management and the sorting of the data a computer software package, NVivo was used. Validation was secured from the research team after the data was analysed in order to verify the researcher's interpretation of the data.

**Table 1. Focus group topic guide.**

| |
|---|
| **Aim: To explore the experiences of undergraduate student nurses in a university setting, of using mobile spaced learning as a digital learning platform for symptom management education.** |
| • Would you share any previous experience you have had of using a digital learning platform? |
| • What was your experience of using the digital learning platform? |
| • What aspects of the digital learning platform were easy to use? |
| • What aspects of the digital learning platform were more challenging to use? |
| • What are your opinions of spaced learning being used to enhance your knowledge retention in university education? |
| • How was the digital learning platform a useful resource with regards enhancing your learning? |
| • What aspects of the digital learning platform were enjoyable? |
| • How could the digital learning platform be improved? |
| • What were your perceptions about digital learning before it started? |
| • How did your perceptions of digital learning change as it progressed? |
| • How do you think this has enhanced your learning? |
| • How do you feel this helped your learning? |
| • What positive experiences did you take away from the experience? |
| • What particular, did you feel could have been different? |
| • What were your thoughts and feelings regarding the use of technology in this way? |
| • Would you like to see this type of approach used in other topics? |
| **Anything else you would like to add to the discussion which we have not covered? Thank you.** |

### Ensuring rigour

To enhance the trustworthiness of the qualitative component of this study, it is essential that rigour is demonstrated throughout the study [14]. One method used to demonstrate credibility was the use of reflective questioning. The researcher was reflexive when interpreting the participants' responses, throughout the focus group, in an attempt to capture the meaning the participants were hoping to convey. Transferability of the results generated in the study was achieved through 'thick description' [14]. This entailed the researcher providing explicit accounts of the experiences of the participants' and not just detailing the surface phenomena of their interpretations but also uncovering the meaning behind their feelings and action [15]. An audit trail was maintained to promote dependability and confirmability [14].

## Results

### Survey results

Questions in the survey linked to several areas as participants were asked to answer questions in relation to the usability and acceptability of the spaced learning platform using a Likert scale (Table 2). All Participants (n = 12) expressed that spaced learning was a good medium for education activity and that the education activities were engaging. The majority of participants (91.67%) expressed that the spaced learning platform was interactive, and all participants enjoyed using the learning platform. All participants felt that the spaced learning provided information that helped them to develop their learning in relation to holistic symptom management and utilised a digital platform that was easily navigated. All participants indicated that the spaced learning helped them to translate knowledge from teaching into practice, more than traditional teaching methodologies. All felt that this intervention should be part of the undergraduate nursing curriculum. When asked what they enjoyed the most about the learning platform, the majority of participants expressed that they found the instant feedback very informative and also felt this enhanced their learning. However, when asked what they enjoyed least about the spaced learning many suggested that the digital platform used was confusing and hard to navigate.

Participants made several free text suggestions (Table 3) in relation to how this spaced learning platform could be enhanced for future students. Participants felt that a future digital platform would benefit from increased functionality that could be accessed through a mobile application. Participants also suggested that there should be a distinct correlation between the learning outcomes of the module and the information maintained within the learning platform. Results also illuminated that students felt this would be a complementary addition to traditional didactic teaching methodologies within the nursing curriculum.

### Focus group findings

Findings displayed that students had previous experience using digital learning platforms and felt that they enhanced their learning and engagement in class. An element of digital learning that students particularly found beneficial was the ability to be provided with immediate feedback. As exampled by a quote from student S15, with the spaced learning platform learning was enhanced by the way feedback was delivered and formatted:

> "*You are never sure if you get the answer right in other means but because you were getting the answer straight away and the additional feedback and the additional resource if you got it wrong then you had the opportunity to go and find out why.*"

**Table 2. Quantitative survey results.**

| Question | Strongly Agree | Agree | Disagree | Strongly Disagree |
|---|---|---|---|---|
| **Please indicate your level of agreement with the following usability questions:** | | | | |
| 1. Technical difficulties significantly reduced my ability to participate | 0 | 8.33% (n = 1) | 58.33% (n = 7) | 33.33% (n = 4) |
| 2. This was a good medium for education activity | 83.33% (n = 10) | 16.67% (n = 2) | 0 | 0 |
| 3. This tool allowed me to access educational content easily | 66.67% (n = 8) | 33.33% (n = 4) | 0 | 0 |
| 4. The educational activities were engaging | 91.67% (n = 11) | 8.33% (n = 1) | 0 | 0 |
| 5. The educational activities were interactive | 75.00% (n = 9) | 16.67% (n = 2) | 8.33% (n = 1) | 0 |
| 6. Sufficient time was available for me to answer the questions | 91.67% (n = 11) | 8.33% (n = 1) | 0 | 0 |
| 7. There were too many questions for each case scenario | 0 | 0 | 33.33% (n = 4) | 66.67% (n = 8) |
| 8. The case scenarios were easy to understand | 83.33% (n = 10) | 8.33% (n = 1) | 8.33% (n = 1) | 0 |
| 9. The questions were easy to understand | 91.67% (n = 11) | 8.33% (n = 1) | 0 | 0 |
| 10. I enjoyed using the learning platform | 83.33% (n = 10) | 16.67% (n = 2) | 0 | 0 |
| 11. I engaged with all the case scenarios I was invited to | 75.00% (n = 9) | 16.67% (n = 2) | 8.33% (n = 1) | 0 |
| **Did the learning tool. . .** | | | | |
| 12. Avoid the use of unnecessary jargon? | 50.00% (n = 6) | 41.67% (n = 5) | 8.33% (n = 1) | 0 |
| 13. Use simple language and short sentences? | 58.33% (n = 7) | 33.33% (n = 4) | 8.33% (n = 1) | 0 |
| 14. Contain instructions that were clear and unambiguous? | 75.00% (n = 9) | 25% (n = 3) | 0 | 0 |
| 15. Provide information that developed your learning on symptom management? | 83.33% (n = 10) | 16.67% (n = 2) | 0 | 0 |
| 16. Contain documents and pages that followed simple and consistent formats? | 66.67% (n = 8) | 33.33% (n = 4) | 0 | 0 |
| 17. Contain no blinking, flashing, or sparkling animated images? | 83.33% (n = 10) | 16.67% (n = 2) | 0 | 0 |
| 18. Provide enough information to easily use the learning platform and engage with the learning? | 58.33% (n = 7) | 33.33% (n = 4) | 8.33% (n = 1) | 0 |
| **I feel that the learning experience. . .** | | | | |
| 19. Offered an opportunity to personally learn through participating | 91.67% (n = 11) | 8.33% (n = 1) | 0 | 0 |
| 20. Was enjoyable | 75.00% (n = 9) | 25% (n = 3) | 0 | 0 |
| 21. Has improved my skills as a nursing student | 50.00% | 50.00% | 0 | 0 |
| 22. Helps translate knowledge from teaching into practice more than other teaching sessions you have been involved in | 66.67% (n = 8) | 33.33% (n = 4) | 0 | 0 |
| 23. Would be something I would recommend to other students undertaking the BSc Hons Nursing Programme | 83.33% (n = 10) | 16.67% (n = 2) | 0 | 0 |
| 24. Provided useful and relevant feedback to enhance my learning | 83.33% (n = 10) | 16.67% (n = 2) | 0 | 0 |
| **Please indicate your level of agreement with the following statements:** | | | | |
| 25. I would like to continue to participate in digital learning platforms if the opportunity arose in the future | 83.33% (n = 10) | 16.67% (n = 2) | 0 | 0 |
| 26. The learning activity was a useful resource for developing my knowledge of symptom management | 83.33% (n = 10) | 16.67% (n = 2) | 0 | 0 |

**Table 3. Free text comments.**

| Question | Sample of Free Text Comments |
|---|---|
| What aspect of the experience did you enjoy most? | *"Interactive and engaging learning."* |
| | *"The information provided after each answer to help enhance understanding."* |
| | *"It was interactive and the ability to complete the questions at my own leisure."* |
| | *"I liked the instant feedback and the fact the answer contained good rationale, which was easily followed. The links to external sources for further reading were also valuable. I like the fact the scenarios themselves applied the holistic principles when looking at the patient, it was not just about e.g. the pain level and level of analgesia required, there were a lot of details about the person, which reinforces the person centred spirit in nursing, that we are being taught."* |
| What aspect of the experience did you enjoy least? | *"No reminders were sent so I forgot one assignment."* |
| | *"The [learning] platform took a bit of effort to navigate via mobile phone, especially when looking for the latest quiz. Not a major problem but this is a wrinkle which could be ironed out in my opinion."* |
| | *"Navigating the platform–not very clear."* |
| Please provide other comments or clarification on your above answers. | *"The platform took a while to get use to but after a few times is was easy. It was also good to be able to complete the questions while on the go."* |
| | *"I get app/mobile typing fatigue and if a platform is challenging to use I am likely to disengage."* |
| | *"I think this style of learning can be applied to any subject to reinforce the lectures content and help to retain a bit more knowledge and show students how it is applied in real life cases."* |
| What recommendations would you make for future education using this learning platform? | *"App based and send reminders."* |
| | *"I'd recommend perhaps altering the case scenarios each 'round' so that you don't just memorise the correct answer and complete the quiz by rote. I found myself doing this on the second and third rounds rather than critically thinking about the scenario. Overall though I think this is very beneficial initiative and enjoyed the problem-solving experience of applying knowledge to patient cases—it reinforced my learning of the lecture material, gave a taste of applying this knowledge in actual nursing practice, showed me where my knowledge gaps were and gave me a confidence boost when I got the right answers."* |

However, students highlighted the importance of having a user friendly digital platform to deliver the spaced learning as they found the platform difficult to navigate, as discussed by student S3:

*"I personally thought it was hard to navigate around. I was getting an email saying your quiz today and when I went to do it, I couldn't figure out which one it was."*

Overall findings from the focus group suggested that students felt they would have benefited from having access to the spaced learning through a mobile application. This was also highlighted as being necessary to ensure students received relevant alerts in relation to completing the spaced learning. Student S2 shared the challenges in remembering to participate in the spaced learning if not accessing alerts:

"*The likes of me I don't have my student email sent to my phone so there was actually that I completely forgot all about it and not getting any alerts...*"

Findings displayed that students perceived that the spaced learning allowed for greater knowledge retention which enhanced the learning they gained from the digital platform. This is demonstrated by a quote from student S22:

"*It (spaced learning) is good; I think it makes it stick in your head.*"

However, one participant (S2) felt that this may have been related to remembering the ordering of the answer and not the learning itself:

"*I don't know if I just remember the answers as opposed to the answer. If the question had of been different... I was looking at it 'oh that's that one before I even read it...*"

Students additionally conveyed that although they enjoyed digital spaced learning, they felt it was useful alongside traditional teaching methodologies, rather than seeing it as a replacement. This is displayed by a quote from student S3:

"*Obviously you can't ever replace the traditional style of learning with it, but I think it's a lovely compliment that goes along well with it.*"

## Discussion

Overall, findings conveyed that students viewed digital spaced learning as an acceptable learning platform that enhanced their learning. Similarly to previous research [2, 5], participants expressed that this teaching methodology enhanced their ability to transfer knowledge into clinical practice which is key to nurse education [16]. Findings additionally illuminated that the learning platform delivered optimal and effective feedback which is an important component in effective education delivery [17]. Students indicated that the spaced learning activities improved their engagement with the education material [17], this may be due to the platform encouraging an active learning approach [18]. Within the university setting, traditional teaching methodologies, such as didactic lectures, are often viewed as being suboptimal in achieving effective engagement with students [19]. Conversely, findings from this study have demonstrated that students feel a blended approach to learning [20] is important as they viewed digital learning as complimentary to traditional modes of education delivery. Previous research has highlighted that mobile spaced education can result in longer term knowledge retention [4]. Although the present study was small scale and did not employ outcome measures, students perceived that this method of learning had the potential to increase knowledge retention. However, participants identified the need for a more interactive and adaptive digital platform for future spaced learning initiatives. Previous research has employed the use of the digital microlearning application Qstream [1–5]. This may offer an acceptable platform for future delivery of this teaching innovation to enhance the acceptability and usability of the spaced learning intervention. Future interventional research could consider the use of this more accessible digital platform that is compatible with mobile devices, and provides alerts to users to enhance engagement with the material.

### Limitations

This was a small pilot study with a low response rate of 13%. This may have limited the findings as they are not generalizable to the whole cohort. Additionally, specific outcome measures

were not employed to explore the knowledge gained by students after completion of the spaced learning. However, this was a small acceptability study that has generated some useful insights into how this teaching methodology could be effectively explored in the future.

## Conclusions

In conclusion, findings from the survey suggests that implementation of a digital spaced learning intervention would be acceptable to undergraduate nursing students learning about holistic symptom management. The results of this study provide important insights regarding how this innovative teaching methodology may be optimally delivered using a mobile digital platform. Further interventional research is needed to explore the implementation of this educational intervention using an enhanced digital microlearning platform. The impact of this intervention on long-term knowledge retention amongst UG nursing students would also be an interesting extension of the study.

## Supporting information

**S1 File. Survey.**
(PDF)

## Acknowledgments

The authors wish to thank all the participants who took the time to participate in this study.

## Author Contributions

**Conceptualization:** Clare Mc Veigh, Susan Carlisle, Matt Birch, Helen Kerr.

**Data curation:** Clare Mc Veigh, Christine Oliver.

**Formal analysis:** Clare Mc Veigh, Susan Carlisle, Matt Birch, Lindsay Ace, Christine Oliver, Helen Kerr.

**Funding acquisition:** Clare Mc Veigh.

**Investigation:** Clare Mc Veigh.

**Methodology:** Clare Mc Veigh.

**Project administration:** Clare Mc Veigh.

**Resources:** Clare Mc Veigh.

**Software:** Clare Mc Veigh, Matt Birch.

**Supervision:** Clare Mc Veigh.

**Validation:** Clare Mc Veigh.

**Writing – original draft:** Clare Mc Veigh, Lindsay Ace, Christine Oliver, Helen Kerr.

**Writing – review & editing:** Clare Mc Veigh, Susan Carlisle, Matt Birch, Lindsay Ace, Christine Oliver, Helen Kerr.

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
