## [Decision Letter · Decision Letter 0]

31 Jan 2022

PONE-D-21-35186Pilot study to explore the use of mobile SPaced LEarning as a digital learning platform when teaching symptom management to undergraduate Nursing Students: SPLENdidS studyPLOS ONE

Dear Dr. Mc Veigh,

Thank you for submitting your manuscript to PLOS ONE. After careful consideration, we feel that it has merit but does not fully meet PLOS ONE’s publication criteria as it currently stands. Therefore, we invite you to submit a revised version of the manuscript that addresses the points raised during the review process.

We look forward to receiving your revised manuscript.

Kind regards,

Prabhat Mittal, Ph.D.

Academic Editor

PLOS ONE

Journal Requirements:

2. ‘Please include your tables as part of your main manuscript and remove the individual files. Please note that supplementary tables (should remain/ be uploaded) as separate "supporting information" files.

“This research is funded by the Martha Mc Menamin Memorial Scholarship.”

Please note that funding information should not appear in other areas of your manuscript. We will only publish funding information present in the Funding Statement section of the online submission form.

“CV received the funding from the Martha Mc Menamin Memorial fund. Funders did not play a role in the study design, data collection and analysis, decision to publish, or preparation of the manuscript”

6. Please include a copy of Table 2 and 3 which you refer to in your text on page 10

Reviewers' comments:

Reviewer's Responses to Questions

**Comments to the Author**

1. Is the manuscript technically sound, and do the data support the conclusions?

Reviewer #1: Yes

Reviewer #2: Yes

2. Has the statistical analysis been performed appropriately and rigorously? 

Reviewer #1: Yes

Reviewer #2: Yes

3. Have the authors made all data underlying the findings in their manuscript fully available?

Reviewer #1: Yes

Reviewer #2: Yes

4. Is the manuscript presented in an intelligible fashion and written in standard English?

Reviewer #1: Yes

Reviewer #2: Yes

5. Review Comments to the Author

Reviewer #1: Overall it is an interesting study. The manuscript is well written and structured. The mixed methods approach (case based and MCQs) used for the pilot study seems appropriate. However, some of the observation are as follows:

1. The authors have referred to some of the previous studies in the context. However, some more review of literature could be added to support the current pilot study.

2. The authors have only considered the first year students of the undergraduate nursing course. Why were the other year students not taken into consideration? This would have enabled the authors to get a better response rate (13% seems too low) and hence the study could have been more useful in drawing conclusions about spaced learning.

3. It would have been a more conclusive study if the authors would have added comments/solutions to some of the challenges faced in spaced learning quoted in the manuscript as comments from the participating students.

For example: “The likes of me I don’t have my student email sent to my phone so there was actually

that I completely forgot all about it and not getting any alerts…”

“I don’t know if I just remember the answers as opposed to the answer. If the question

had of been different… I was looking at it ‘oh that’s that one before I even read it…”

Reviewer #2: This is an interesting study. The paper is well-written and organized in general. Look for a few grammatical mistakes. Include a few recommendations based on your findings. The discussion and conclusion sections can be a little more in-depth.

6. PLOS authors have the option to publish the peer review history of their article (what does this mean?). If published, this will include your full peer review and any attached files.

Reviewer #1: No

Reviewer #2: No

---

## [Author Response · Author response to Decision Letter 0]

13 Apr 2022

Reviewers' comments:

Reviewer #1

Overall it is an interesting study. The manuscript is well written and structured. The mixed methods approach (case based and MCQs) used for the pilot study seems appropriate. However, some of the observation are as follows:

1. The authors have referred to some of the previous studies in the context. However, some more review of literature could be added to support the current pilot study.

Thank you for highlighting this. This has now been added to the introduction section on pages 3 and 4. 

2. The authors have only considered the first year students of the undergraduate nursing course. Why were the other year students not taken into consideration? This would have enabled the authors to get a better response rate (13% seems too low) and hence the study could have been more useful in drawing conclusions about spaced learning.

Thank you for this observation, this is explained on Page 7 when we justify the use of convenience sampling. 

3. It would have been a more conclusive study if the authors would have added comments/solutions to some of the challenges faced in spaced learning quoted in the manuscript as comments from the participating students.

For example: “The likes of me I don’t have my student email sent to my phone so there was actually

that I completely forgot all about it and not getting any alerts…”

“I don’t know if I just remember the answers as opposed to the answer. If the question

had of been different… I was looking at it ‘oh that’s that one before I even read it…”

Thank you for highlighting this. This has now been added to the discussion and conclusion on page 17. 

Reviewer #2: This is an interesting study. The paper is well-written and organized in general. Look for a few grammatical mistakes. Include a few recommendations based on your findings. The discussion and conclusion sections can be a little more in-depth.

Thank you for highlighting this. This has now been added to the discussion and conclusion on page 17.

---

## [Editor Report · Decision Letter 1]

25 May 2022

Pilot study to explore the use of mobile SPaced LEarning as a digital learning platform when teaching symptom management to undergraduate Nursing Students: SPLENdidS study

PONE-D-21-35186R1

Dear Dr. Mc Veigh,

We’re pleased to inform you that your manuscript has been judged scientifically suitable for publication and will be formally accepted for publication once it meets all outstanding technical requirements.

Kind regards,

Prabhat Mittal, Ph.D.

Academic Editor

PLOS ONE

Additional Editor Comments (optional):

I have revisited the manuscript and reviews with all attachments (tables and figures). Based on received reviews, author response and in my opinion, authors have appropriately discussed the use of quantitative and qualitative analysis (thematic approach) to meet the objectives of the manuscript. The analysis has been carried out based on a survey of undergraduate nursing students about the mobile spaced learning as a digital platform. Study recruitments (sample and participants) have also been appropriately justified in the manuscript. Table 2 and table 3 sufficiently discusses the results of the descriptive statistics and the summary of focused group discussion outcome of thematic analysis
---

## [Editor Report · Acceptance letter]

30 May 2022

PONE-D-21-35186R1 

Pilot study to explore the use of mobile SPaced LEarning as a digital learning platform when teaching symptom management to undergraduate Nursing Students: SPLENdidS study 

Dear Dr. Mc Veigh:

I'm pleased to inform you that your manuscript has been deemed suitable for publication in PLOS ONE. Congratulations! Your manuscript is now with our production department. 

Kind regards, 

on behalf of

Dr. Prabhat Mittal 

Academic Editor

PLOS ONE